# Bioguided Identification of Active Antimicrobial Compounds from *Asphodelus bento-rainhae* and *Asphodelus macrocarpus* Root Tubers

**DOI:** 10.3390/ph16060830

**Published:** 2023-06-01

**Authors:** Maryam Malmir, Katelene Lima, Sérgio Póvoas Camões, Vera Manageiro, Maria Paula Duarte, Joana Paiva Miranda, Rita Serrano, Isabel Moreira da Silva, Beatriz Silva Lima, Manuela Caniça, Olga Silva

**Affiliations:** 1Research Institute for Medicines (iMed.ULisboa), Faculty of Pharmacy, Universidade de Lisboa, 1649-003 Lisbon, Portugal; m.malmir@campus.ul.pt (M.M.); k.lima@campus.ul.pt (K.L.); sergiocamoes@campus.ul.pt (S.P.C.); jmiranda@ff.ulisboa.pt (J.P.M.); rserrano@campus.ul.pt (R.S.); icsilva@ff.ulisboa.pt (I.M.d.S.); beatrizlima@netcabo.pt (B.S.L.); 2National Reference Laboratory of Antibiotic Resistances and Healthcare-Associated Infections, Department of Infectious Diseases, National Institute of Health Dr. Ricardo Jorge, 1649-016 Lisbon, Portugal; vera.manageiro@insa.min-saude.pt (V.M.); manuela.canica@insa.min-saude.pt (M.C.); 3Centre for Animal Science Studies (CECA), Institute of Agricultural and Agro-Food Sciences and Technologies (ICETA), University of Porto, 4050-453 Porto, Portugal; 4Associate Laboratory for Animal and Veterinary Sciences (AL4AnimalS), Faculty of Veterinary Medicine, Universidade de Lisboa, 1300-477 Lisbon, Portugal; 5The Mechanical Engineering and Resource Sustainability Center (MEtRICs), Nova School of Science and Technology, Universidade Nova de Lisboa, 2829-516 Caparica, Portugal; mpcd@fct.unl.pt; 6Center for Interdisciplinary Research in Animal Health (CIISA), Faculty of Veterinary Medicine, Universidade de Lisboa, 1300-477 Lisbon, Portugal

**Keywords:** antimicrobial activity, anthracene derivatives, *Asphodelus bento-rainhae*, *Asphodelus macrocarpus*, root tubers, toxicity

## Abstract

Root tubers of *Asphodelus bento-rainhae* subsp. *bento-rainhae* (AbR), a vulnerable endemic species, and *Asphodelus macrocarpus* subsp. *macrocarpus* (AmR) have traditionally been used in Portugal to treat inflammatory and infectious skin disorders. The present study aims to evaluate the in vitro antimicrobial activity of crude 70% and 96% hydroethanolic extracts of both medicinal plants, specifically against multidrug-resistant skin-related pathogens, to identify the involved marker secondary metabolites and also to assess the pre-clinical toxicity of these medicinal plant extracts. Bioguided fractionation of the 70% hydroethanolic extracts of both species using solvents of increasing polarity, namely diethyl ether (DEE: AbR-1, AmR-1), ethyl acetate (AbR-2, AmR-2) and aqueous (AbR-3, AmR-3) fractions, enabled the identification of the DEE fractions as the most active against all the tested Gram-positive microorganisms (MIC: 16 to 1000 µg/mL). Furthermore, phytochemical analyses using TLC and LC-UV/DAD-ESI/MS techniques revealed the presence of anthracene derivatives as the main constituents of DEE fractions, and five known compounds, namely 7′-(chrysophanol-4-yl)-chrysophanol-10’-*C*-beta-*D*-xylopyranosyl-anthrone (**p**), 10,7′-bichrysophanol (**q**), chrysophanol (**r**), 10-(chrysophanol-7′-yl)-10-hydroxychrysophanol-9-anthrone (**s**) and asphodelin (**t**)**,** were identified as the main marker compounds. All these compounds showed high antimicrobial activity, particularly against *Staphylococcus epidermidis* (MIC: 3.2 to 100 µg/mL). Importantly, no cytotoxicity against HepG2 and HaCaT cells (up to 125 µg/mL) for crude extracts of both species and genotoxicity (up to 5000 µg/mL, with and without metabolic activation) for AbR 96% hydroethanolic extract was detected using the MTT and Ames tests, respectively. Overall, the obtained results contribute to the concrete validation of the use of these medicinal plants as potential sources of antimicrobial agents in the treatment of skin diseases.

## 1. Introduction

Antimicrobial resistance (AMR) is a growing global healthcare problem due to the loss of efficacy of first-line antibiotics. Many pathogens are developing resistance to multiple drugs, making infections difficult or, in some cases, impossible to treat [1]. In response to the increasing demand for alternative medicines, the screening of natural products has emerged as one of the most successful methods for detecting/identifying antibacterial agents. Although in recent decades, the majority of new antibacterial drugs were from natural sources [2], only a small fraction of marine, fungal and plant resources have been investigated, and nature still offers a high potential for drug-lead discovery, notably among anti-infective compounds [3].

In this context, ethnomedical knowledge has an important role in plant-derived drug discovery [4,5], and based on this information, the genus *Asphodelus* L. belonging to the family *Asphodelaceae* is referred to as one of the most promising sources of medicinal plants [6]. Root tubers of *Asphodelus* species have been traditionally used for the treatment of skin-related disorders and infections such as wounds, eczema, alopecia and psoriasis [6]. Furthermore, ethnomedical information on several *Asphodelus* species, supported by in vitro and in vivo biological activity studies, indicates their strong antimicrobial potential [7,8,9,10,11,12,13], particularly against resistant pathogens, due to the presence of secondary metabolites such as anthraquinones, arylcoumarins, terpenoids and naphthalene derivatives [12,14,15,16,17,18,19,20,21,22]. 

The Portuguese flora exhibits a considerable abundance of *Asphodelus* species, sub-species and varieties compared to the rest of Europe and the Mediterranean Basin. Besides the above-mentioned medical applications, root tubers are also used as daily food in the Iberian Peninsula, after being moistened and fried to eliminate the astringent compounds [6].

*Asphodelus bento-rainhae* subsp. *bento-rainhae* P. Silva is a vulnerable [23] endemic species from the Gardunha mountain range [24], located in the central region of Portugal, co-existing with *Asphodelus macrocarpus* subsp. *macrocarpus* Parlatore in the same geographical area. Both species are commonly known by the Portuguese name “abrotea”, and their root tubers have traditionally been used for the treatment of skin diseases such as scabies, dermatophytosis and warts in Portugal.

The objective of this study is derived from the fact that although there are promising ethnomedical, phytochemical and biological data related to the *Asphodelus* species, to the best of our knowledge, no scientific studies on *Asphodelus bento-rainhae* and *Asphodelus macrocarpus* root tubers have been documented so far. Thus, the present study aims to establish the chemical profiles of the potential antimicrobial constituents together with the pre-clinical safety evaluation and validation of the use of the studied plants as herbal medicines.

## 2. Results and Discussion

### 2.1. Drug–Extract Ratio (DRE)

The drug–extract ratio was calculated as 1.9:1 and 5.5:1 for the *A. bento-rainhae* root tuber (AbR) and 2.7:1 and 6.7:1 for the *A. macrocarpus* root tuber (AmR) 70% and 96% hydroethanolic extracts, respectively. Considering these results, AbR exhibited a higher percentage of yield in both hydroethanolic extracts compared to the AmR extracts. Moreover, extraction with ethanol 96% noticeably reduced the percentage of yield in both species. 

### 2.2. Bioguided Phytochemical Analysis

#### 2.2.1. Phytochemical Screening and Antimicrobial Activity

Hyphenated analytical techniques were applied for the phytochemical dereplication of the samples. Following our previous study’s results [25], the chromatographic profiles of AbR and AmR extracts showed excellent qualitative similarity in their chemical composition, characterized by the presence of terpenoids, phenolic acids and anthracene derivatives. Therefore, in continuation of the above-mentioned study searching for potent antimicrobial metabolites from these Portuguese *Asphodelus* species, liquid–liquid fractionations of both plant extracts with increasingly polar solvents, namely diethyl ether (AbR-1, AmR-1), ethyl acetate (AbR-2, AmR-2) and water (AbR-3, AmR-3), were performed. Both species’ crude extracts and their subsequent L-L fractions were then submitted to in vitro antimicrobial evaluation in order to select the most active fractions for further phytochemical identification of their lead secondary metabolites. 

The antimicrobial activity of the crude extracts, their derived L-L fractions and isolated compounds were evaluated through determination of the MIC values, an in vitro quantitative method of susceptibility testing against both selected Gram-positive and Gram-negative resistant pathogens.

As shown in Table 1, among all the tested samples, AbR-1 and AmR-1 were the only fractions that demonstrated prominent antibacterial activity against all the Gram-positive microorganisms (*Staphylococcus* spp. strains), with MIC values ranging from 16 to 1000 µg/mL. The AbR-1 fraction exhibited stronger antibacterial activity, with about two-fold higher inhibition potential against *S. saprophyticus* INSA842, *S. epidermidis* INSA958, *S. epidermidis* INSA960, *S. haemolyticus* INSA982 and *S. haemolyticus* INSA984 and about three-fold stronger activity against *S. epidermidis* INSA796 when compared to the AmR-1 fraction. No activity (MIC > 2000 µg/mL) was found against Gram-negative microorganisms (*Escherichia coli*, *Klebsiella pneumoniae*, *Pseudomonas aeruginosa* and *Acinetobacter baumannii*) in the tested range of concentrations (up to 2000 µg/mL). 

Considering the previously reported results of the antimicrobial activity of *Asphodelus* spp. root tuber crude extracts and in agreement with the obtained results verified in our species, weak to moderate activities against a similar pathogen panel with MIC values higher than 2000 µg/mL were observed [8]. The methanolic root extracts of *A. luteus* and *A. microcarpus* showed antimicrobial potential against methicillin-resistant *S. aureus* (MRSA), with MIC values of 650 to 1250 and 1250 to 2500 µg/mL, respectively [7]. Screening *A. microcarpus* tuber methanolic extract using an agar well diffusion assay revealed moderate activity against *S. aureus,* with an inhibition diameter zone of 14 mm [26]. Furthermore, the 80% hydromethanolic whole plant extract of *A. tenuifolius* was also found to be significantly active against *S. aureus*, *E. coli*, *P. aeruginosa* and *K. pneumonia,* with inhibition diameter zones of 16, 29, 18 and 18 mm, respectively, determined using the disc diffusion method [11,13].

#### 2.2.2. Isolation, Detection and Tentative Identification of the Main Bioactive Marker Compounds

Following the obtained antimicrobial activity results, the AbR-1 and AmR-1 fractions were selected for further phytochemical characterization of their constituents. LC-UV/DAD-ESI/MS spectral data (Figure 1) confirmed the presence of phenolic acids, phenylpropanoids, anthracene derivatives and triterpenoids in both fractions, and six compounds, namely chlorogenic acid (**b**, *t*_R_: 11.7 min, λ_max_: 241sh, 296sh, 326 nm), vanillic acid (**c**, *t*_R_: 12.4 min, λ_max_: 261, 292sh nm), caffeic acid (**d**, *t*_R_: 12.8 min, λ_max_: 240sh, 296sh, 324 nm), ferulic acid (**g**, *t*_R_: 18.5 min, λ_max_: 235sh, 296sh, 323 nm), isochlorogenic acid A (**h**, *t*_R_: 20.4 min, λ_max_: 242sh, 296sh, 326 nm) and *β*-sitosterol (**u**, R*_f_*: 0.4, λ_max_: 248 nm), were identified based on co-chromatography techniques using authentic standards. In order to separate, purify and identify the major marker constituents of both extracts and active fractions, further phytochemical studies were conducted. 

The AbR-1 fraction was submitted to the column chromatography technique using reversed-phase Silica gel 90 C_18_ and Sephadex LH-20. The procedure resulted in the isolation of five anthraquinones in pure form (Figure 2). 

Compound **p** was identified as 7′-(chrysophanol-4-yl)-chrysophanol-10′-*C*-beta-*D*-xylopyranosyl-anthrone, also known as 10′R)-1,1′,8,8′-tetrahydroxy-10′-beta-*D*-xylopyranosyl-3,3′-dimethyl-4,7′-bianthracene-9,9′,10(10′H)-trione, with empirical formula C_35_H_28_O_11_ and PubChem CID:102153707, found as a reddish amorphous powder, with UV: λ_max_ (CH_3_CN) 255, 288sh, 366, 428 nm and mass of 625 [M + H]^+^ and fragments of 493 and 475 (*m/z*). Compound **q** was identified as 10,7′-bichrysophanol (chrysalodin), with empirical formula C_30_H_20_O_9_ and PubChem CID:13940829, found as orange crystals, with UV: λ_max_ (CH_3_CN) 262, 292sh, 388, 432 nm, mass of 523 [M-H]^−^ and prominent fragments of 269 and 253 (*m/z*). Compound **r** was identified as chrysophanol (chrysophanic acid), with empirical formula C_15_H_10_O_4_ and PubChem CID:10208, found as orange crystals, with UV: λ_max_ (CH_3_CN) 256, 288sh, 429 nm and mass of 253 [M-H]^−^ (*m*/*z*). Compound **s** was identified as 10-(chrysophanol-7′-yl)-10-hydroxychrysophanol-9-anthrone, with empirical formula C_30_H_20_O_8_ and PubChem CID:14584824 found as orange crystals, with UV: λ_max_ (CH_3_CN) 262, 292sh, 388, 434 nm and mass of 507 [M-H]^−^ and prominent fragments of 490 and 253 (*m/z*). Compound **t** was identified as asphodelin (4,7′-bichrysophanol), with empirical formula C_30_H_18_O_8_ and PubChem CID: 182665, found as dark red crystals with UV: λ_max_ (CH_3_CN) 260, 290sh, 434 nm and mass of 505 [M-H]^−^ and fragments of 488, 460 and 253 (*m/z*).

All the isolated compounds have been previously identified in *Asphodelus* species. For instance, chlorogenic acid, caffeic acid and vanillic acid were reported from an *Asphodelus ramosus* L. whole plant extract [27]. Chlorogenic acid was detected in the leaf extract of *Asphodelus aestivus* Brot. [28] and caffeic acid was reported from the flower extract of *A. ramosus* [29]. 7′-(chrysophanol-4-yl)-chrysophanol-10′-*C*-beta-*D*-xylopyranosyl-anthrone was also reported from the root tuber extract of *A. ramosus* [30]. 10,7′-bichrysophanol was identified as one of the main anthracene derivatives of *A. acaulis*, *A. albus* and *A. fistulosus* root extracts [14]. This compound was also reported from seed extracts of *A. microcarpus* [31]. 

Chrysophanol was identified as the major marker compound of the diethyl ether fraction of both *A. bento-rainhae* and *A. macrocarpus* root tuber extracts. This compound was found to be the most common anthracene derivative of the *Asphodelus* species, reported from *A. acaulis*, *A. albus*, *A. fistulosus* [14] and *A. microcarpus* [19] root and *A. albus* [32,33], *A. fistulosus* [17,34], *A. macrocarpus* subsp. *rubescens* [32] and *A. microcarpus* aerial part extracts [17,35]. 10-(chrysophanol-7′-yl)-10-hydroxychrysophanol-9-anthrone was recorded from *A. microcarpus* [19] and *A. ramosus* [36] root and leaf [35] extracts. Asphodelin was found in *A. acaulis*, *A. albus* and *A. fistulosus* root extracts [14]. This compound was also detected in *A. microcarpus* root [18,19] and seed [31] extracts. 

Since anthracene derivatives are considered the main secondary metabolites of *Asphodelus* species, the detected and identified compounds could effectively be used for the chemotaxonomic classification of both *A. bento-rainhae* and *A. macrocarpus* species [14,37]. 

*β*-sitosterol, a common phytosterol, was identified in root extracts of *A. albus*, *A. microcarpus* and *A. tenuifolius* [15,38,39,40] and seeds extracts of *A. fistulosus* and *A. microcarpus* [41].

#### 2.2.3. Antimicrobial Activity of the Major Marker Compounds and 96% Hydroethanolic Extracts of Both Asphodelus Root Tubers

The results for the antimicrobial activity of the five isolated major marker compounds of both diethyl ether L-L fractions (AbR-1, AmR-1) are presented in Table 2. Additionally, considering the chemical class and polarity of these compounds and in order to verify whether the activity of the total extract is relevant to the major or minor constituents, a less polar hydroethanolic extract (96%) of both species was also prepared and tested. 

The AbR 96% hydroethanolic extract was found to be the most active crude extract and showed higher contents of marker metabolites in comparison to AmR 96% and both species’ 70% hydroethanolic extracts, which is in accordance with the fundamental role of these compounds in the antimicrobial activity exhibited by these medicinal plants. 

All the tested compounds were found to be active against all the tested Gram-positive strains, particularly against *Staphylococcus epidermidis,* with MIC values between 3.2 and 100 µg/mL (Table 2). Among these strains, teicoplanin- and linezolid-resistant *S. epidermidis* INSA958 showed the highest susceptibility to all the tested compounds. Moreover, chrysophanol, the major marker compound of both species, showed remarkable activity (MIC: 3.2 µg/mL) against this strain, which is often found on the human skin and mucous membrane; however, according to hospital surveillance reports, the bacterium is a common cause of nosocomial wound infections. Similar to the obtained results of the tested crude extracts, no activity regarding these compounds was found against the tested Gram-negative microorganisms in the tested range of concentrations (up to 200 µg/mL).

To the best of our knowledge, no data related to the resistant strains employed in this study have been reported; however, chrysophanol and its derivatives have been previously reported to have potential antibacterial activity against other *S. aureus* strains (MIC values of 90 to 190 µg/mL) [42]. 

So far, there has not been enough research to explain the antibacterial mechanism of these compounds; however, according to the existing studies, the cell walls of Gram-positive bacteria, compared to Gram-negative bacteria, are more sensitive to many antibiotics and antimicrobial chemical compounds/herbal drugs [43]. The lipopolysaccharide layer and periplasmic space of Gram-negative bacteria are the reasons for the relative resistance of Gram-negative bacteria [44].

### 2.3. Pre-Clinical Safety Assessment

#### 2.3.1. Evaluation of the Cytotoxicity Potential

The results of the in vitro cytotoxicity evaluations of the *A. bento-rainhae* (AbR) 70% and 96% hydroethanolic extracts are presented in Figure 3. The analysis of these data obtained through the cell viability assay clearly showed that none of the tested extracts induced cytotoxicity in HepG2 cells. However, since this medicinal plant is commonly used for the treatment of skin disorders, we further assessed cytotoxicity using a skin cell type. For this, the AbR 96% hydroethanolic extract, as the most active extract with the highest contents of marker secondary metabolites, and its major constituent, chrysophanol, were selected. 

As previously observed with HepG2, the AbR 96% extract did not reduce HaCaT viability, indicating its safe use through topical application at concentrations up to 125 µg/mL, but the same was not observed with chrysophanol, which reduced cell viability by up to 50% with concentrations higher than 25 µg/mL. 

#### 2.3.2. Evaluation of the Genotoxicity/Mutagenicity Potential

Although the negative results of the genotoxic/mutagenic potential of the root tuber 70% hydroethanolic extracts of both species were previously reported by the authors [25], as suggested by the guidelines [45], a genotoxicity assessment of different herbal preparations should be evaluated in order to reflect, as far as possible, the full spectrum of the extracted components. Additionally, since the AbR 96% hydroethanolic extract exhibited the highest antimicrobial activity and quantity of the active secondary metabolites, it was selected for further genotoxicity/mutagenicity evaluations. 

The obtained results of the Ames test for the AbR 96% hydroethanolic extract are presented in Table 3. According to the genotoxicity guidelines [46,47], a mutagenic substance in the bacterial reverse mutation (Ames) test should exhibit a reproducible dose-related increase in the number of revertant colonies for at least one of the tester strains. Additionally, the number of revertant colonies must be more than twice the number of colonies produced on the negative (solvent) control plates. For cytotoxicity, a reduction in the number of revertants and/or clearing or diminution of the background lawn might be detected [48,49,50].

The analysis of the results showed that none of the tested concentrations of this extract (up to 5000 µg/plate) enhanced the number of revertant colonies in any tested strains with or without metabolic activation compared to the negative control. Moreover, toxicity did not occur, since none of the above-mentioned requirements occurred at any tested concentration. Therefore, under the conditions of this study, the mutagenic potential essential to ensure the safety of these extracts was not observed. 

Even though there are studies indicating the genotoxic potential of chrysophanol in the Ames test with metabolic activation (S9) [51], the obtained negative results of the tested AbR 96% hydroethanolic crude extract (with and without metabolic activation), show that the presence of chrysophanol does not influence the genotoxicity of the crude extract. Insufficient amounts of the mutagenic constituents and their interactions in the extracts/complex mixtures are among the various theories that could explain this phenomenon. Additionally, human exposure to chrysophanol and its derivatives through AbR 96% hydroethanolic extract is expected to be negligible, concerning the expected mode of administration (topical application), since they need to undergo bioactivation, mediated by different isoforms of cytochrome P 450, to become genotoxic [52].

## 3. Methods and Materials

### 3.1. Chemical and Biological Reagents

2-aminoanthracene, 9-aminoacridine hydrochloride monohydrate, ammonium sodium phosphate dibasic tetrahydrate, benzo(*a*)pyrene, chlorogenic acid, chrysophanol, *d*-(+)-biotin, dimethyl sulfoxide/DMSO, glucose monohydrate, glucose-6-phosphate, nicotinamide adenine dinucleotide phosphate (NADP^+^), 2-nitrofluorene and *tert*-butyl hydroperoxide/T-BHP were obtained from Sigma-Aldrich (St. Louis, MO, USA). Anisaldehyde, l-histidine monohydrochloride monohydrate, magnesium sulfate heptahydrate, methanol and potassium hydroxide were purchased from Merck (Darmstadt, Germany). Caffeic acid, ferulic acid and vanillic acid were acquired from Extrasynthese (Genay, France). Citric acid monohydrate, di-sodium hydrogen phosphate dihydrate and sodium dihydrogen phosphate monohydrate were purchased from PanReac AppliChem (Barcelona, Spain). Sodium chloride and di-potassium hydrogen phosphate were from Honeywell Fluka™ (Seelze, Germany). β-sitosterol and 2-aminoethyl diphenylborinate were obtained from Acros organics (Geel, Belgium). Bacto™ agar was acquired from Becton Dickinson & Co (Franklin Lakes, NJ, USA), ethanol was sourced from Carlo Erba Reagents (Val-de-Reuil, France), glacial acetic acid came from Chem-Lab NV (Zedelgem, Belgium), polyethylene glycol 400/PEG was sourced from VWR Chemicals (Rosny-sous-Bois, France), sulfuric acid was acquired from PanReac AppliChem (Barcelona, Spain), sodium azide came from J.T. Baker Chemical Company (Phillipsburg, NJ, USA) and nutrient broth (NB) Nº 2 was sourced from Oxoid (Basingstoke, UK). Aroclor 1254-induced rat liver S9 was purchased from Trinova Biochem (GmbH, Giessen, Germany). In preparing all solutions, dilutions and culture media, ultra-pure water from a Milli-Q water purification system, Millipore (Molsheim, France), was used.

### 3.2. Plant Materials

Root tubers of *A. bento-rainhae* (AbR) and *A. macrocarpus* (AmR) were collected from Serra da Gardunha, Portugal, during root dormancy in November 2019. The corresponding voucher specimens were deposited in the Laboratory of Pharmacognosy, Department of Pharmacy, Pharmacology and Health Technologies, Faculty of Pharmacy, Universidade de Lisboa (voucher specimens: OSilva_201901-*A. bento rainhae* and OSilva_201902-*A. macrocarpus*). The collected samples were dried in a well-ventilated dark space at room temperature. The authors’ previous monographic study give a more detailed description of both species’ botanical identification and sample selections [25]. 

### 3.3. Preparation of Extracts

The collected samples were dried and extracted using the maceration method (with a mixture of ethanol/water 70% and 96%) at room temperature under agitation (3×, 24 h each). Hydroethanolic extracts were concentrated under reduced pressure using a rotary evaporator and freeze-dried. The obtained AbR and AmR 70% hydroethanolic extracts were then submitted to liquid–liquid partitioning (L-L), generating the diethyl ether (AbR-1, AmR-1), ethyl acetate (AbR-2, AmR-2) and aqueous (AbR-3, AmR-3) fractions.

### 3.4. Chromatographic Analysis

Thin-layer chromatography (TLC) was performed using Silica gel 60 F_254_ and RP-18 F_254_ pre-coated plates (Merck^®^, Darmstadt, Germany). Anisaldehyde–sulfuric acid, natural product polyethylene glycol reagent (NP/PEG = NEU) and potassium hydroxide (KOH) 5% ethanolic solution [53] were used as spray reagents for the detection of the secondary marker metabolites such as terpenoids, phenolic acids and anthracene derivatives, respectively.

High-performance liquid chromatography (HPLC) was carried out using a Waters Alliance 2690 Separations Module coupled with a Waters 996 photodiode array detector (UV/DAD) (Waters Corporation, Milford, MA, USA). Crude extracts (20 mg/mL) and L-L fractions (10 mg/mL) were initially solubilized in acetonitrile/water, and standard solutions (1 mg/mL) were prepared in acetonitrile and filtered through a polytetrafluoroethylene syringe filter (0.2 µm). An Atlantis RP-18 T3 column (5 µm, 150 × 4.6 mm) was used for the analysis of 25 µL of the injected samples with a flow rate of 1 mL/min. Water with 0.1% (*v/v*) formic acid (solvent A) and acetonitrile (solvent B) were used as the mobile phase, and gradients of 95% A: 5% B to 0% A: 100% B for a total run time of 75 min were used. Chromatograms were monitored and registered on *Maxplot* (wavelength 240–650 nm), and the obtained data were analyzed using Waters Millennium^®^ 32 Chromatography Manager Software (Waters Corporation, Milford, MA, USA). 

Mass spectrometry (MS) analysis was conducted using the same HPLC equipment in tandem with a triple quadrupole mass spectrometer (Micromass^®^ Quatro Micro^TM^ API, Waters^®^, Drinagh, Ireland) using an electrospray ionization source (ESI) operating in both positive and negative mode. Data were acquired and analyzed using MassLynx™ V4.1 software (Waters^®^, Drinagh, Ireland).

### 3.5. Isolation and Identification of the Main Marker Compounds

One gram of the active extract (AbR-1) was applied to the Sephadex LH-20 column. Several fractions were collected and concentrated through evaporation of the solvent. Then, the TLC control of the fractions was performed on silica gel 60 RP-18 plates using an H_2_O: MeOH (0.5:19.5, *v/v*) solvent system and screened under UV_254_ and UV_366_. Fractions with similar profiles were mixed, and the collected fractions were bulked into six main fractions: AbR-1a (665 mg), AbR-1b (60 mg), AbR-1c (255 mg), AbR-1d (117 mg), AbR-1e (71 mg) and AbR-1f (29 mg). Compounds **p**, **q**, **r**, **s** and **t** were purified using a C18 reversed-phase silica gel column eluted with MeOH: H_2_O (90:10). The identification of compounds was based on co-chromatographic techniques and the obtained data related to the retention times, ultraviolet absorption and mass spectral characteristics recorded using LC-UV/DAD-ESI/MS, together with their TLC characteristics in comparison to those of standards and published data.

### 3.6. In Vitro Antimicrobial Activity

The broth microdilution method was used for an in vitro evaluation of the antibacterial potential [54], using 96-well tissue culture plates (VWR^®^, Radnor, PA, USA) to determine the minimum inhibitory concentrations (MIC) of the tested samples against twelve reference (ATCC, LGC Standards S.L.U., Barcelona, Spain) and clinical strains (INSA clinical strains collection) of both Gram-positive (*Staphylococcus aureus*, *S. epidermidis*, *S. saprophyticus*, *S. haemolyticus*) (Table 4) and Gram-negative (*Escherichia coli*, *Klebsiella pneumoniae*, *Pseudomonas aeruginosa*, *Acinetobacter baumannii*) multidrug-resistant bacteria. 

The samples to be tested were initially prepared in water or DMSO 10%, and serial dilutions (2–2000 μg/mL for crude extracts/fractions and 0.2–200 for pure compound) were performed in a Mueller–Hinton medium and were distributed (50 μL) in each of the microplate wells using a microplate liquid handler (Precision^TM^ BioTek, Winooski, VT, USA).

Inoculums were prepared from a pure bacterial culture on agar, and suspensions with a turbidity of 0.5 for Gram-negative and 0.25 for Gram-positive bacteria on the McFarland scale (Grant Bio™ DEN-1B, Cambridge, UK) were prepared in Mueller–Hinton medium and stored at 4 °C until use. For MIC determination, the prepared suspensions were diluted at a ratio of 1:10, and 50 µL of this dilution was added to all the wells. To verify the absence of contamination and to check the viability of the inoculum, two controls were included for each tested sample, one plate in the absence of the extract solution and the other in the presence of the solvent (DMSO). As previously described, all experiments were carried out in triplicate to obtain consistent values.

### 3.7. In Vitro Cytotoxicity Evaluation using MTT Assay

In vitro cytotoxicity evaluation was performed using the methylthiazolyldiphenyl-tetrazolium bromide (MTT) reduction assay [55] on the human liver cell line HepG2 (ATCC, Cat. No. HB-8065, Manassas, VA, USA) and the human spontaneously immortalized keratinocyte cell line HaCaT (CLS, Cat. No. 300493, Eppelheim, Germany). HepG2 and HaCaT were inoculated at a density of 8.5 × 10^4^ cells/cm^2^ in α-MEM (Sigma-Aldrich^®^, St. Louis, MO, USA) with 1 mM sodium pyruvate (PAN Biotech, Aidenbach, Germany), 1% non-essential amino acids (NEAA; PAN Biotech) and 10% fetal bovine serum (FBS, Gibco^®^ Thermo Fisher Scientific^TM^ (Waltham, MA, USA), and of 4.0 × 10^4^ cells/cm^2^ in DMEM (Sigma-Aldrich^®^) with 4 g/L *D*-(+)-glucose (AppliChem, Darmstadt, Germany) and 10% fetal bovine serum (FBS, Gibco^®^ Thermo Fisher Scientific^TM^ (Waltham, MA, USA), respectively. Both cell lines were maintained in a humidified chamber at 37 °C in a 5% CO_2_ atmosphere. After 48 h, the medium was replaced with fresh medium with AbR 70% and 96% extracts and chrysophanol (9:1) at final concentrations of 25, 50, 75, 100 and 125 µg/mL for 48 h. Complete cell culture medium, DMSO 1% and DMSO 20% in α-MEM or DMEM were used as a positive, solvent and negative control, respectively. After cell washing with PBS, 200 μL 0.5 mg/mL MTT (Sigma-Aldrich^®^) was added to the cell culture medium. HepG2 and HaCaT were incubated for 3 h and 2 h, respectively, in a humidified chamber at 37 °C in a 5% CO_2_ atmosphere. Next, 200 μL DMSO was used for solubilizing the purple crystals formed prior to measuring absorbance at 570 nm using a microplate spectrophotometer (SPECTROstar Omega; BMG LabTech, Ortengerg, Germany). The results are expressed as a percentage relative to the solvent control. Four wells were used for each sample, and at least two independent experiments were performed.

Data analysis and graphs were plotted using GraphPad Prism^®^ software (version 9.0.0.121, GraphPad Software, San Diego, CA, USA). The results are presented as mean ± standard deviation. *p* < 0.05 was considered significant.

### 3.8. In Vitro Genotoxicity/Mutagenicity Evaluation using Ames Test

The screening of the genotoxicity potential was performed using a bacterial reverse mutation test (the Ames test) for the detection of genotoxic carcinogens and relevant genetic changes. The technique was conducted following the OECD No. 471 [46] and ICH S2 (R1) [47] guidelines as well as the published reference protocols [56,57]. 

*Salmonella enterica* serovar Typhimurium tester strains (TA98, TA100, TA102, TA1535 and TA1537) were used in this study (with and without metabolic activation) in a direct plate incorporation method. TA100, TA98, TA102 and TA1535 were kindly provided by the Genetic Department of the Nova Medical School of the Universidade NOVA de Lisboa (Portugal), having received them from Professor B.N. Ames (Berkeley, CA, USA). TA1537 was obtained from ATCC, NUMBER: 29630™, LOT: 7405375.

S9 mix (10%, *v/v* rat liver S9, 0.4 M MgCl_2_, 1.65 M KCl, 1 M glucose-6-phosphate, 0.1 M nicotinamide adenine dinucleotide phosphate and 0.2 M sodium phosphate buffer, pH 7.4) was freshly prepared and kept on ice throughout the experiment.

The AbR 96% hydroethanolic extract (50 mg/mL) was initially dissolved in DMSO, and 100 µL of the extract dilutions was mixed with 500 µL sodium phosphate buffer (0.1 M, pH 7.4) (in the assay without metabolic activation) or S9 mix (in the assay with metabolic activation). Then, 100 µL of the bacterial culture and 2 mL of melted top-agar, supplemented with 0.05 mM biotin and histidine, were added to the mixture. After a 48 h incubation at 37 °C, manual counting of His+ revertant colonies for each concentration was performed. The results are expressed as the mean number of revertant colonies with the standard deviation (mean ± SD). The positive controls were sodium azide (SA, 1.5 µg/plate for TA100 and TA1535), 2-nitrofluorene (2-NF, 5 µg/plate for TA98), 9-aminoacridine (9-AA, 100 µg/plate for TA1537) and tert-butyl hydroperoxide (tBHP, 50 µg/plate for TA102) in the assay without metabolic activation, and 2-aminoathracene (2-AA, 2 µg/plate for TA98 and 10 µg/plate for TA102, TA1535 and TA1537) and benzo(*a*)pyrene (BaP, 5 µg/plate for TA100) in the assay with metabolic activation. All assays were performed in triplicate to obtain consistent values. 

## 4. Conclusions

Overall, the observed antimicrobial activity of both the *A. bento-rainhae* and *A. macrocarpus* root tuber 70% hydroethanolic extracts were similar to those obtained and reported from the other *Asphodelus* spp. tested against a similar panel of pathogens. However, the fractionation of these extracts and an enriched 96% hydroethanolic extract certainly enhanced their significant antimicrobial activity, as they contain the highest amounts of 1,8-dihydroxy anthracene derivatives, a known chemical class of secondary metabolites with potential antimicrobial activity. 

Moreover, the isolated and identified chrysophanol derivatives could be considered important chemotaxonomic markers of both studied *Asphodelus* species. 

## Figures and Tables

**Figure 1 pharmaceuticals-16-00830-f001:**
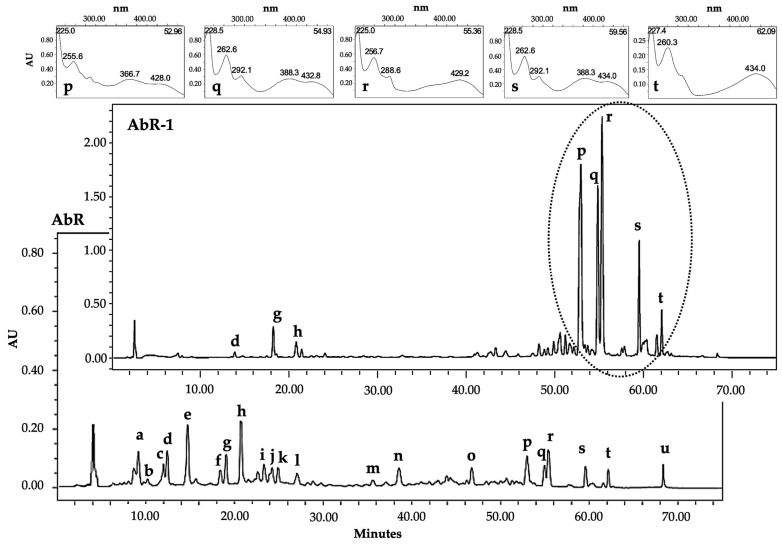
Comparative LC-UV/DAD *Maxplot* (210–600 nm) chromatographic profiles of *A. bento-rainhae* root tuber extract and its highly active diethyl ether L-L fraction. Abbreviations: AbR, *A. bento-rainhae* root tuber extract; AbR-1, AbR diethyl ether fraction; Major isolated compounds (**p** to **t**) and their UV spectra.

**Figure 2 pharmaceuticals-16-00830-f002:**
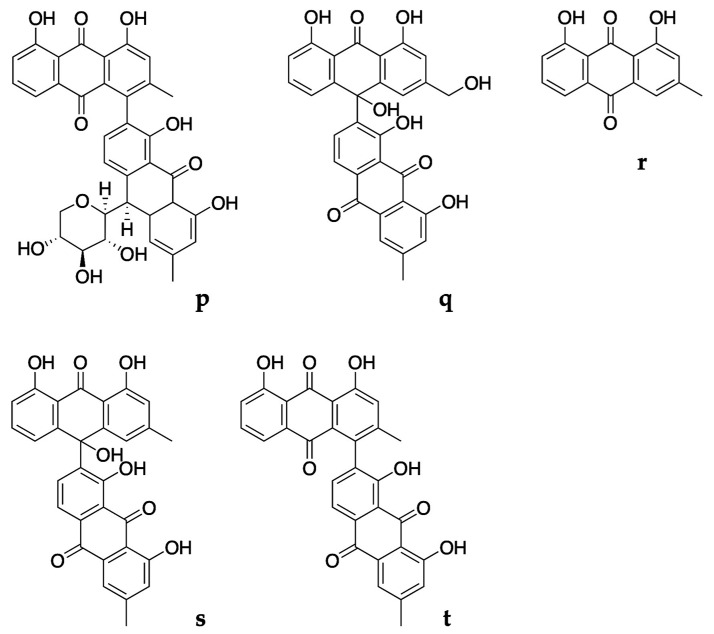
Major marker compounds of *A. bento-rainhae* and *A. macrocarpus* root tuber extracts. Abbreviations: **p**, 7′-(chrysophanol-4-yl)-chrysophanol-10′-*C*-beta-*D*-xylopyranosyl-anthrone; **q**, 10,7′-bichrysophanol; **r**, chrysophanol; **s**, 10-(chrysophanol-7′-yl)-10-hydroxychrysophanol-9-anthrone; **t**, asphodelin.

**Figure 3 pharmaceuticals-16-00830-f003:**
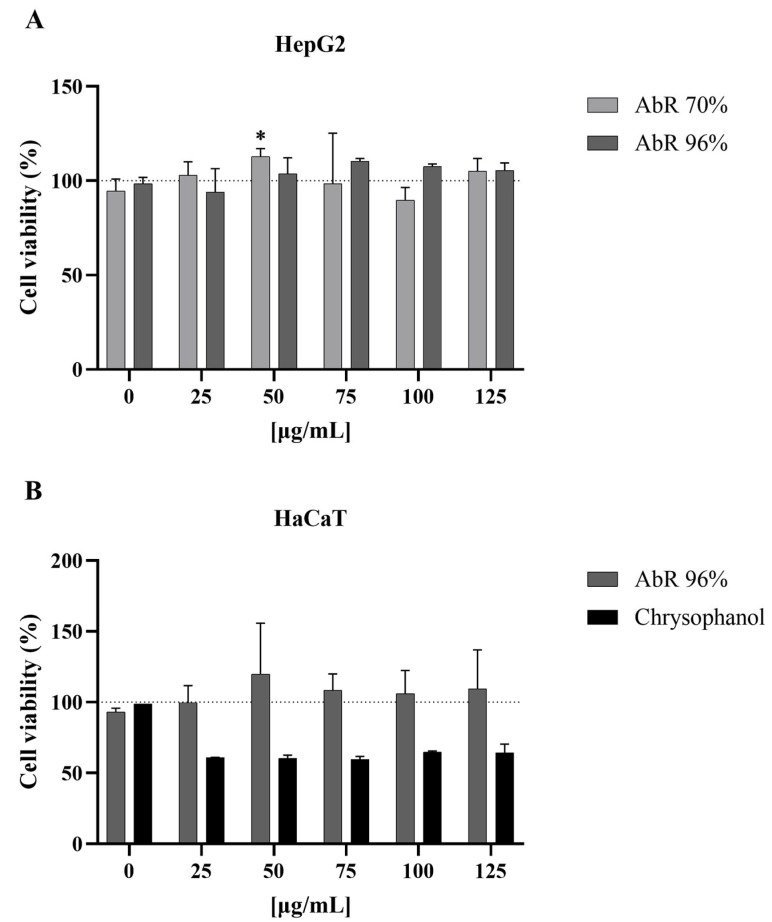
HepG2 (**A**) and HaCaT (**B**) cell viability after 48 h of incubation with *A. bento-rainhae* root tuber 70% and 96% hydroethanolic extracts and chrysophanol, evaluated using methylthiazolyldiphenyl-tetrazolium bromide (MTT) reduction assay. Data are shown as the percentage of solvent control (dashed line) and mean ± standard deviation; *n* = 2–5. * *p* < 0.05.

**Table 1 pharmaceuticals-16-00830-t001:** In vitro antimicrobial activity of crude extracts and their subsequent L-L fractions of *A. bento-rainhae* and *A. macrocarpus* root tubers against Gram-positive strains.

Bacteria (Gram +)	MIC (µg/mL)							
	AbR 70%	AmR 70%	AbR-1	AmR-1	AbR-2	AmR-2	AbR-3	AmR-3
*S. aureus* ATCC 29213	>2000	>2000	250	250	>2000	>2000	>2000	>2000
*S. aureus* CQINSA4923	>2000	>2000	125	125	>2000	>2000	>2000	>2000
*S. aureus* INSA790	>2000	>2000	250	250	>2000	>2000	>2000	>2000
*S. aureus* INSA936	>2000	>2000	250	250	>2000	>2000	>2000	>2000
*S. aureus* INSA896	>2000	>2000	125	125	>2000	>2000	>2000	>2000
*S. saprophyticus* INSA842	>2000	>2000	125	250	>2000	>2000	>2000	>2000
*S. saprophyticus* INSA867	>2000	>2000	500	1000	>2000	>2000	>2000	>2000
*S. epidermidis* INSA796	>2000	>2000	125	500	>2000	>2000	>2000	>2000
*S. epidermidis* INSA958	>2000	>2000	250	500	>2000	>2000	>2000	>2000
*S. epidermidis* INSA960	>2000	>2000	125	250	>2000	>2000	>2000	>2000
*S. haemolyticus* INSA982	>2000	>2000	16	32	>2000	>2000	>2000	>2000
*S. haemolyticus* INSA984	>2000	>2000	62	125	>2000	>2000	>2000	>2000

Abbreviations: AbR 70%, *A. bento-rainhae* root tuber 70% hydroethanolic extract; AmR 70%, *A. macrocarpus* root tuber 70% hydroethanolic extract; AbR-1, AbR diethyl ether fraction; AbR-2, AbR ethyl acetate fraction; AbR-3, AbR aqueous fraction; AmR-1, AmR diethyl ether fraction; AmR-2, AmR ethyl acetate fraction; AmR-3, AmR aqueous fraction. ATCC, American Type Culture Collection; INSA, Instituto Nacional de Saúde clinical strains collection; MIC, minimum inhibitory concentration.

**Table 2 pharmaceuticals-16-00830-t002:** In vitro antimicrobial activity of 96% hydroethanolic extracts and the isolated marker compounds (**p** to **t**) of *A. bento-rainhae* and *A. macrocarpus* root tuber extracts against Gram-positive strains.

Bacteria (Gram +)	MIC (µg/mL)						
	AbR 96%	AmR 96%	p	q	r	s	t
*S. aureus* ATCC 29213	125	1000	25	100	100	100	200
*S. aureus* CQINSA4923	125	2000	100	100	200	100	100
*S. aureus* INSA790	500	>2000	100	200	200	200	200
*S. aureus* INSA936	250	>2000	100	200	200	200	200
*S. aureus* INSA896	250	>2000	100	200	200	200	200
*S. saprophyticus* INSA842	500	>2000	200	200	200	200	200
*S. saprophyticus* INSA867	1000	>2000	200	200	200	200	200
*S. epidermidis* INSA796	500	>2000	25	100	50	100	50
*S. epidermidis* INSA958	1000	2000	12.5	12.5	3.2	12.5	100
*S. epidermidis* INSA960	250	>2000	12.5	100	100	100	100
*S. haemolyticus* INSA982	125	2000	6.25	200	100	200	100
*S. haemolyticus* INSA984	250	>2000	6.25	200	200	200	200

Abbreviations: AbR 96%, *A. bento-rainhae* root tuber 96% hydroethanolic extract; AmR 96%, *A. macrocarpus* root tuber 96% hydroethanolic extract. ATCC, American Type Culture Collection; INSA, Instituto Nacional de Saúde clinical strains collection; MIC, minimum inhibitory concentration. **p**, 7′-(chrysophanol-4-yl)-chrysophanol-10′-*C*-beta-*D*-xylopyranosyl-anthrone; **q**, 10,7′-bichrysophanol; **r**, chrysophanol; **s**, 10-(chrysophanol-7′-yl)-10-hydroxychrysophanol-9-anthrone; **t**, asphodelin.

**Table 3 pharmaceuticals-16-00830-t003:** Mutagenicity of *A. bento-rainhae* 96% hydroethanolic extract in the bacterial reverse mutation test (Ames test).

AbR 96% µg/Plate	Number of Revertant Colonies without Metabolic Activation, Mean (*n* = 3) ± Standard Deviation (SD)				
	TA98	TA100	TA102	TA1535	TA1537
500	39 ± 2	150 ± 3	319 ± 8	15 ± 5	17 ± 2
1000	37 ± 5	166 ± 8	306 ± 9	16 ± 3	18 ± 6
2000	38 ± 1	142 ± 6	305 ± 12	13 ± 3	22 ± 6
2500	42 ± 4	152 ± 9	301 ± 4	10 ± 1	24 ± 8
3750	45 ± 1	163 ± 8	327 ± 3	13 ± 3	24 ± 3
5000	49 ± 3	143 ± 11	320 ± 8	16 ± 1	20 ± 6
NC	38 ± 6	142 ± 2	320 ± 4	15 ± 2	20 ± 2
PC	2-NF	SA	tBHP	SA	9-AA
	488 ± 30	1048 ± 43	881 ± 26	827 ± 13	1354 ± 5
	Number of Revertant Colonies with Metabolic Activation (S9), Mean (*n* = 3) ± Standard Deviation (SD)				
1000	43 ± 1	145 ± 1	221 ± 6	12 ± 4	11 ± 4
2000	33 ± 3	147 ± 1	217 ± 5	12 ± 6	13 ± 2
4000	33 ± 4	162 ± 2	215 ± 5	11 ± 6	11 ± 3
5000	36 ± 1	159 ± 6	237 ± 2	13 ± 3	14 ± 1
NC	44 ± 8	157 ± 6	172 ± 2	11 ± 2	12 ± 1
PC	2-AA	BaP	2-AA	2-AA	2-AA
	832 ± 35	947 ± 148	732 ± 12	266 ± 1	306 ± 50

Abbreviations: AbR 96%, *A. bento-rainhae* root tuber 96% hydroethanolic extract; Nd, not determined; NC, negative control/solvent control (DMSO 30%); PC, positive control reference; 2-NF, 2-nitrofluorene; SA, sodium azide; *t*BHP, *tert*-butyl hydroperoxide; 9-AA, 9-aminoacridine; 2-AA, 2-aminoathracene; BaP, benzo(*a*)pyrene.

**Table 4 pharmaceuticals-16-00830-t004:** Composition of the Gram-positive pathogen panel under study.

Bacteria (Gram +)			Demonstration of Resistance to the Antibiotics									
	CIP	DAP	ERY	FA	FOX	GN	LNZ	OXA	PEN	TEI	TET	VAN
*S. aureus* ATCC 29213								S	MS			
*S. aureus* CQINSA4923	R		R	S	R	R	S	R	R	S	S	S
*S. aureus* INSArefV					R					R		R
*S. aureus* INSA936		R										
*S. aureus* INSA896	R			R	R		R					
*S. saprophyticus* INSA842			R	R								
*S. saprophyticus* INSA867										R		
*S. epidermidis* INSA796	R				R		R			R		
*S. epidermidis* INSA958							R			R		
*S. epidermidis* INSA960										R		
*S. haemolyticus* INSA982	R				R					R		
*S. haemolyticus* INSA984	R	R			R							

Abbreviations: ATCC: American Type Culture Collection, INSA: Instituto Nacional de Saúde clinical strains collection, CIP: ciprofloxacin, DAP: daptomycin, ERY: erythromycin, FA: fusidic acid, FOX: cefoxitin, GN: gentamicin, LNZ: linezolid, OXA: oxacillin, PEN: penicillin, TEI: teicoplanin, TET: tetracycline, VAN: vancomycin, MS: methicillin-susceptible, R: resistant, S: susceptible.

## Data Availability

Data is contained within the article.

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
