# Peer review of "Bioguided Identification of Active Antimicrobial Compounds from Asphodelus bento-rainhae and Asphodelus macrocarpus Root Tubers"

_pharmaceuticals, 2023, doi:10.3390/ph16060830_

Round 1

Author Response

Reviewer comments: Please check the attached file.

Authors response: Dear reviewer, thank you very much for your comments. We have attentively addressed all the comments in the revised version of the manuscript as follows:

Point 1. In the line 27, Check the spelling of the word.

Authors response 1: the spelling was checked and corrected accordingly.

Point 2. In the lines 28 to 31, It is difficult to understand the meaning because too many commas are used in the sentence.

Authors response 2: In the “Abstract” section, to clarify the meaning, the referred paragraph (new lines of 27 to 31) was modified as follows:

“Bio-guided fractionation of the 70% hydroethanolic extracts of both species using solvents of increasing polarity, namely, diethyl ether (DEE: AbR-1, AmR-1), ethyl acetate (AbR-2, AmR-2) and aqueous (AbR-3, AmR-3) fractions enabled the identification of the DEE fractions, as the most active, against all the tested Gram-positive microorganisms (MIC: 16 to 1000 µg/mL).”

Point 3. In the lines 37 to 38, Basic information about the cytotoxicity and genotoxicity evaluation methods should be given in the abstract. e.g., cell types and/or method name should be mentioned.

Authors response 3: In the “Abstract” section, brief information about the cytotoxicity and genotoxicity evaluation methods (new lines of 38 to 40), were added to the revised manuscript as follows:

“Importantly, no cytotoxicity against HepG2 and HaCaT cells (up to 125 µg/mL) for crude extracts of both species and genotoxicity (up to 5000 µg/mL, with and without metabolic activation) for AbR 96% hydroethanolic extract was detected using MTT and Ames tests, respectively.”

Point 4. In the line 101, Please use the term "antibacterial", as you only evaluated the antimicrobial activity against "bacteria".

Authors response 4: The word “antimicrobial” was replaced by “antibacterial”.

Point 5. In the line 105 to 107, This is an important outcome of the present study and should be discussed in the Discussion part.

Authors response 5: In the section “Results and Discussion”, the following paragraphs (new lines of 239 to 241 and 246 to 251), were added to the revised manuscript.

“Similar to the obtained results of the tested crude extracts, no activity regarding these compounds was found against the tested Gram-negative microorganisms in the tested range of concentrations (up to 200 µg/mL).”

“So far, there has not been enough research to explain the antibacterial mechanism of these compounds; however, according to the studies, the cell walls of gram-positive bacteria, compared to gram-negative bacteria, are more sensitive to many antibiotics and antimicrobial chemical compounds /herbal drugs [43]. The lipopolysaccharides layer and periplasmic space of gram-negative bacteria are the reasons for the relative resistance of gram-negative bacteria [44].”

Point 6. In the line 124 to 126, As "agar diffusion method" and "microdilution method" are different, the authors should emphasize the testing methods of the references 11, 13 and 26.

Authors response 6: In the section “Results and Discussion” the missing data was added to the revised manuscript and the following paragraph (new lines of 133 to 138) was modified as follows:

“Screening of A. microcarpus tubers methanolic extract using agar well diffusion assay revealed moderate activity against S. aureus with an inhibition diameter zone of 14 mm [26]. 80% hydromethanolic whole plant extract of A. tenuifolius was also found to be significantly active against S. aureus, E. coli, P. aeruginosa, and K. pneumonia with inhibition diameter zones of 16, 29, 18 and 18 mm, respectively, determined by disc diffusion method [11,13]”

Point 7. In the line 414, The references should be listed in an order as they used in the text. Please check the "Instructions for Authors" to number the References.

Authors response 7: Dear reviewer, all the references were checked and placed correctly in the revised manuscript.

Point 8. In the lines 447 to 449, As these referenced comments are not the results of the present study, they cannot be used to conclude the findings. These references might be moved to the Results and/or Discussion parts.

Authors response 8: In the section “Results and Discussion”, the following paragraph (lines 205 to 207), was added to the manuscript.

“Since anthracene derivatives are considered the main secondary metabolites of Asphodelus species, the detected and identified compounds could effectively be used for the chemotaxonomic classification of both A. bento-rainhae and A. macrocarpus species [14,37].”

Reviewer 2 Report

Author should revisit their own presentation of the paper before acceptance of the manuscript.

Status: major revision

1. Plant scientific names should be properly written with author citations of nomenclature

2. in the introductions in depth ethno-medicinal values should be discussed which trigger the authors to work on these plants.

3. Introduction section is very weak in presentation. Authors can go through the following papers to improve 

Antimicrobial activity of select edible plants from Odisha, India against food-borne pathogens. LWT-Food Science and Technology 113:108246. doi: 10.1016/j.lwt.2019.06.013

Large Scale Screening of Ethnomedicinal Plants for Identification of Potential Antibacterial Compounds. Molecules21(3): 293; doi:10.3390/molecules21030293

3. Authors must present some pictorial proof of activities of the plant extracts against microbes

4. Material methods : need more clear explanations of the methods authors followed.

5. Give more highly resolution images.

6. Many typo errors , please go through each sentences.

7. Authors are requested to do some in silico work for the identified compounds with specific bacterial proteins to get some concrete ideas on their binding potentials.

English editing is required after revision

Author Response

Reviewer comments: “Author should revisit their own presentation of the paper before acceptance of the manuscript. Status: major revision”.

Authors response: Dear reviewer, thank you very much for your comments. We have attentively addressed all the comments in the revised version of the manuscript as follows:

Reviewer comments:

Point 1. Plant scientific names should be properly written with author citations of nomenclature.

Authors response 1: Dear reviewer, we absolutely agree with your comment, the scientific name of each species cited for the first time, is completely spelled out in the manuscript (e.g., Asphodelus bento-rainhae subsp. bento-rainhae P. Silva). For the second time, the common scientific name (e.g., Asphodelus bento-rainhae) and afterward, an abbreviation of the genus (e.g., A. bento-rainhae) is mentioned in the manuscript.

Point 2. In the introductions in depth ethno-medicinal values should be discussed which trigger the authors to work on these plants.

Point 3. Introduction section is very weak in presentation. Authors can go through the following papers to improve.

- Antimicrobial activity of select edible plants from Odisha, India against food-borne pathogens. LWT-Food Science and Technology 113:108246. doi: 10.1016/j.lwt.2019.06.013

- Large Scale Screening of Ethnomedicinal Plants for Identification of Potential Antibacterial Compounds. Molecules21(3): 293; doi:10.3390/molecules21030293

Authors response 2,3: Dear reviewer, a detailed description of the genus and the selected species and their in-depth ethnomedicinal values has been previously published by the authors through several publications:

  • Malmir, M.; Serrano, R.; Caniça, M.; Silva-Lima, B.; Silva, O. A Comprehensive Review on the Medicinal Plants from the Genus Asphodelus. Plants 2018, 7, 20, doi:10.3390/plants7010020.
  • Malmir, M.; Serrano, R.; Lima, K.; Duarte, M.P.; Moreira da Silva, I.; Silva Lima, B.; Caniça, M.; Silva, O. Monographic Quality Parameters and Genotoxicity Assessment of Asphodelus bento-rainhae and Asphodelus macrocarpus Root Tubers as Herbal Medicines. Plants 2022, 11, 3173, doi:10.3390/plants11223173.

However, in order to avoid the self-plagiarism and yet showcasing the concise narrative that highlights the significance of the study, the “Introductions” section was modified as follows:

“Antimicrobial resistance (AMR) is a growing global healthcare problem due to the loss of efficacy of first-line antibiotics. Many pathogens are developing resistance to multiple drugs making infections difficult or in some cases impossible to treat [1]. In response to the increasing demand for alternative medicines, the screening of natural products has emerged as one of the most successful methods for detecting/identifying antibacterial agents. Although in the last decades, the majority of the new antibacterial drugs were from natural sources [2], only a small fraction of marine, fungal and plant resources have been investigated and nature still offers a high potential for drug-lead discovery, notably among anti-infective compounds [3].

In this context, ethnomedical knowledge has an important role in plant-derived drug discovery [4,5] and based on these information, the genus Asphodelus L. belonging to the family Asphodelaceae is referred as one of the most promising source of medicinal plants [6]. Root tubers of Asphodelus species have been traditionally used for the treatment of skin-related disorders and infections such as wounds, eczema, alopecia and psoriasis [6]. Furthermore, the ethnomedical information from several Asphodelus species, supported by in vitro and in vivo biological activity studies, indicates their strong antimicrobial potential [7–13], particularly against resistant pathogens due to the presence of secondary metabolites such as anthraquinones, arylcoumarins, terpenoids and naphthalene derivatives [12,14–22].

The Portuguese flora exhibits a considerable abundance of Asphodelus species, sub-species and varieties within Europe and the Mediterranean Basin. Besides the above-mentioned medical applications, root tubers are also used as daily food in the Iberian Peninsula, after being moistened and fried beforehand to eliminate the astringent compounds [6].

Asphodelus bento-rainhae subsp. bento-rainhae P. Silva is a vulnerable [23] endemic species from the Gardunha mountain range [24], located in the central region of Portugal, co-existing with Asphodelus macrocarpus subsp. macrocarpus Parlatore in the same geographical area. Both species are commonly known by the Portuguese name “abrotea”, and their root tubers have been traditionally used for the treatment of skin diseases such as scabies, dermatophytosis, and warts in Portugal.

The objective of this study lies in the fact that, although there are promising ethnomedical, phytochemical and biological data related to the Asphodelus species, to the best of our knowledge, no scientific studies so far, have been documented on Asphodelus bento-rainhae and Asphodelus macrocarpus root tubers. Thus, the present study aims to establish the chemical profiles of the potential antimicrobial constituents together with the pre-clinical safety evaluation and validation of the use of the studied plants as herbal medicines.”

Point 4. Authors must present some pictorial proof of the activities of the plant extracts against microbes.

Authors response 4: Selective images of the most active extract will be presented in the graphical abstract.

Point 5. Material methods: need more clear explanations of the methods authors followed.

Authors response 5: Dear reviewer, detailed information has been linked through several publications regarding the employed materials and methods followed by authors.

Point 6. Give more highly resolution images.

Authors response 6: Images with higher resolution were placed in the manuscript.

Point 7. Many typo errors, please go through each sentence.

Authors response 7: Several grammatical issues such as spaces, parenthesis, commas and two points, and typo errors have been detected and corrected throughout the revised manuscript.

Point 8. Authors are requested to do some in silico work for the identified compounds with specific bacterial proteins to get some concrete ideas on their binding potentials.

Authors response 8:  Dear reviewer, thanks for your comment and suggestion. The study of the mechanism of action of the identified compounds is already planned for the near future. Since the referred aspect was not among the main objectives of the study, it was not considered essential to include in the present manuscript.

Reviewer 3 Report

Malmir et al. have identified the main marker compounds from the roots of two medicinal plants. The manuscript has been well prepared, but there are several points that need further clarification.

1. How to collect the plant sample and how to identify the type, and the preparation of the extract should be linked to the previous paper Marmir et al., 2022 (https://doi.org/10.3390/plants11223173).

2. Figure 3 seems identical to Table 2. It is better to show only one.

3. On lines 161-162 there seems to be a typo

4. The genotoxicity and mutagenicity of the 70% ethanol extract of the same sample were reported by Marmir et al. 2022 (https://doi.org/10.3390/plants11223173). What is the basic reason that the same test needs to be repeated on 96% ethanol extract? What are the basic differences between the two types of extracts must be discussed.

5. Some literature state that activated chrysophanol is a fundamental mutagenic agent (example: Liberman et al. Appl. Environment. Microb, Sept. 1980, p. 476-479), whereas in the tests reported in this manuscript extracts containing these compounds and some of its derivatives do not show mutagenicity effect. An explanation for the phenomenon needs to be presented in the discussion.

6. In the conclusion section, it is not customary to display the cited literature there because the conclusion must emerge from the results of the experiments conducted.

Author Response

Reviewer comments:

“Malmir et al. have identified the main marker compounds from the roots of two medicinal plants. The manuscript has been well prepared, but there are several points that need further clarification.”

Authors response: Dear reviewer, thank you very much for your comments. We have attentively addressed all the comments in the revised version of the manuscript as follows:

Point 1. How to collect the plant sample and how to identify the type, and the preparation of the extract should be linked to the previous paper Malmir et al., 2022 (https://doi.org/10.3390/plants11223173).

Authors response 1: Dear reviewer, thank you for your suggestion. In the section “Material and Methods” the following information (lines 345 to 346) was linked to the authors' previous study and added to the revised manuscript:

“The authors’ previous monographic study gave a more detailed description of both species’ botanical identifications and sample selections [25]”.

Point 2. Figure 3 seems identical to Table 2. It is better to show only one.

Authors response 2: The referred figure was removed from the manuscript and will be added to the graphical abstract.

Point 3. On lines 161-162 there seems to be a typo

Authors response 3:

Although “Kstmmfcurgnhnw-uhfffaoysa-N seems to be a typo, it is an InChIKey, and in some publications and databases given as a synonym of the “10-(Chrysophanol-7'-Yl)-10-Hydroxychrysophanol-9-Anthrone”.

https://pubchem.ncbi.nlm.nih.gov/compound/14584824

However, to prevent any misinterpretation, it was removed from the manuscript.

Point 4. The genotoxicity and mutagenicity of the 70% ethanol extract of the same sample were reported by Malmir et al. 2022 (https://doi.org/10.3390/plants11223173). What is the basic reason that the same test needs to be repeated on 96% ethanol extract? What are the basic differences between the two types of extracts must be discussed.

Authors response 4: Dear reviewer, thank you for your constructive recommendation. In the section “Results and Discussion”, the following paragraph (lines 274 to 280) was added to the manuscript:

“Although the negative results of the genotoxic/mutagenic potential of the root tuber  70% hydroethanolic extracts of both species were previously reported by the authors [25], however, as suggested by the guidelines [45], genotoxicity assessment of different herbal preparations should be evaluated in order to reflect as far as possible the full spectrum of the extracted components. Additionally, since the AbR 96% hydroethanolic extract exhibited the highest antimicrobial activity and quantity of the active secondary metabolites, it was therefore selected for further genotoxicity/mutagenicity evaluations.” 

Point 5.  Some literature state that activated chrysophanol is a fundamental mutagenic agent (example: Liberman et al. Appl. Environment. Microb, Sept. 1980, p. 476-479), whereas in the tests reported in this manuscript extracts containing these compounds and some of its derivatives do not show mutagenicity effect. An explanation for the phenomenon needs to be presented in the discussion.

Authors response 5: Dear reviewer, thank you for your constructive comment and suggestion. In the section “Results and Discussion”, the following paragraph (lines 302 to 311) was added to the manuscript:

“Even though there are studies indicating the genotoxic potential of chrysophanol in the Ames test with metabolic activation (S9) [51], the obtained negative results of the tested AbR 96% hydroethanolic crude extract (with and without metabolic activation), show that the presence of chrysophanol does not influence the genotoxicity of the crude extract. Insufficient amounts of the mutagenic constituents and their interactions in the extracts/ complex mixtures are among the various theories that could explain this phenomenon. Additionally, the human exposure to chrysophanol and its derivatives through AbR 96% hydroethanolic extract is expected to be negligible, concerning the expected mode of administration (topical application), once  they need to undergo bioactivation, mediated by different isoforms of cytochrome P 450 to become genotoxic [52].”

Point 6.  In the conclusion section, it is not customary to display the cited literature there because the conclusion must emerge from the results of the experiments conducted.

Authors response 6: The cited literatures were removed from the section “Conclusions” and in the section “Results and Discussion”, the following paragraph (lines 205 to 207), was added to the manuscript.

“Since anthracene derivatives are considered the main secondary metabolites of Asphodelus species, the identified and detected compounds could effectively be used for the chemotaxonomic classification of both A. bento-rainhae and A. macrocarpus species [14,37].”

Round 2

Reviewer 2 Report

Though authors have made substantial improvement, but now a days only investigate against some available pathogen has no meaning until a concrete prediction of their mechanism which can only be make these two plants more informative to the future pharmaceutical researchers. So it must be go through in silico evidences of such antimicrobial activities.

Author Response

Dear reviewer, thank you very much for your comments. We agree that in silico studies are informative and useful. However, as indicated in the manuscript, the medicinal plants studied in our work have long been used clinically as traditional herbal preparations for the treatment of skin infectious diseases as shown in the cited reference works. Therefore, in this context, we consider it essential to evaluate these formulations in vitro against the possible microorganisms involved.

This data, along with knowledge of the chemical fingerprint of the secondary metabolites, may allow the sourcing of the elements needed for appropriate quality control and identification of compounds involved in the observed antimicrobial activity. This activity generally relies on a group of compounds acting synergistically (positively and/or negatively) on the same and different targets. This strategy is widely recognized and anchored in the field of ethnopharmacology and thus also in the field of pharmaceutical science. Therefore, as previously mentioned, we are now confident that our results are suitable for independent publication and that the proposed additional in silico work can be realized as study progress in the near future.

Dear reviewer, since we have already revised the introduction, the references and all the parts of the text you quoted in the submitted version, in addition to your comment on the in silico tests, could you please indicate the sentences that now need to be corrected?

Waiting for your help, kind regards

Olga Duarte Silva